# Developing a Quality 4.0 Implementation Framework and Evaluating the Maturity Levels of Industries in Developing Countries

Ali Zulqarnain *, Muhammad Wasif and Syed Amir Iqbal

Department of Industrial & Manufacturing Engineering, NED University of Engineering & Technology, Karachi 75270, Pakistan
* Correspondence: alizul@neduet.edu.pk; Tel.: +92-334-3137166

**Abstract:** Industry 4.0 implementations integrate people, machines, data, technology, and processes and allow organizations to connect through digitization and cloud-based systems. The purpose of the present research is to realize the need and sustainability of digitization and connectivity within the quality management domain in developing countries, which is now called Quality 4.0. In previous research, several Quality 4.0 frameworks have been proposed. However, most of the frameworks are based on philosophy or require vast resources to implement. Hence, this research work proposes a framework for the implementation of Quality 4.0 in different industries. This framework is based on eleven dimensions that are the core requirements of the Quality 4.0; key variables are evaluated to gauge the maturity of the implementation of the framework. A research instrument is developed based on the variables to acquire the industry data, which are statistically analyzed to determine the maturity of implementation. It was found that scalability, culture, and app development require the most immense attention from industry to completely implement the requirements of Quality 4.0. Finally, recommendations are suggested that address the strengths, weaknesses, opportunities, and threats in transforming traditional quality management systems to the Quality 4.0 framework.

**Keywords:** Quality 4.0; Industry 4.0; Quality Sustainability; digital transformation; digitization and connectivity; quality management; developing countries

## 1. Introduction

Defining and implementing advancements in quality are always challenging and daunting. The definition of quality evolved from fitness for use and customer satisfaction to invariability [1]. Acceptable input to the system and output from the system laid the basis for quality control (QC) including seven QC tools that were later revamped as quality assurance (QA) that used seven management tools. QA not only controls the quality of inputs and outputs but also ensures it during processing [1]. The concept of quality engineering was introduced to eliminate the online experimental analysis that caused interruptions to production systems. Quality engineering was introduced to conduct offline experiments for the design of products or the realignment of processes to determine optimal process parameters [2]. Later, the evolution of enterprise-wide quality planning was introduced, which encompasses the strategic alignment of quality with enterprise systems and brought the concepts of total quality management (TQM) and lean six sigma [3].

High demand rates, short product life cycles, and advancements in production technology drive industries to implement high-end automation systems. At one side, these systems have exponentially enhanced throughput rates, but on the other hand, they require robust quality assurance within the production system. In the transformation from Industry 1.0 to Industry 4.0, the need of quality management requirements and implementation of technology increased drastically especially for small and medium enterprises (SME), due to which new tools, techniques, and knowledge were introduced [4–7]. Further innovation

and development are leading industries towards the new era of technology, which is called Industry 5.0 [8,9]. The largest enabler of this industrial revolution is the advancement in technology toward machine-to-machine collaboration. It has not only enhanced the productivity of manufacturing systems but also affected the service industry, e.g., hotels, restaurants, and airlines, as well the health care and education industries [8,9]. The fifth industrial revolution is presently conceptualized for specialized human knowledge and expertise to collaborate with smart systems. The current literature review presents the ideology that Industry 5.0 is going to bring human capital (employees) to industry [8]. It will empower and connect humans with collaborative robots (COBOTS) and other machines and exploit the human expertise and innovation to enhance business process productivity and management by consolidating these processes with the smart systems [10]. Many researchers focused on the role of standardization in adoption to Industry 4.0. For instance, Germany formed a coordination group, and Spain constituted a forum called Standards for Connected Industry 4.0 [4]. Standardization proposes globally common components of information exchange (the same vocabulary, syntax, formats, protocols, management platforms, etc.), design solutions per the standards of Industry 4.0, and specialized training supports for the transformation from traditional ways [4]. With this standardization need, Standardization Council Industrie 4.0 (SCI 4.0), DIN (German Institute for Standardization), DKE (German Commission for Electrical, Electronic & Information Technologies of DIN), and VDE (Verband der Elektrotechnik Elektronik Informationstechnik) formulated a strategic and technically oriented document (road map) in which experts from industry, research, and other relevant fields proposed standards and specifications for successful implementations of Industry 4.0 at local levels and for harmonization at international level [5]. With the transformation of industry, quality management also transformed, which is quite well defined by the American Society of Quality (ASQ) [11]. Figure 1 illustrates the transformation of quality management.

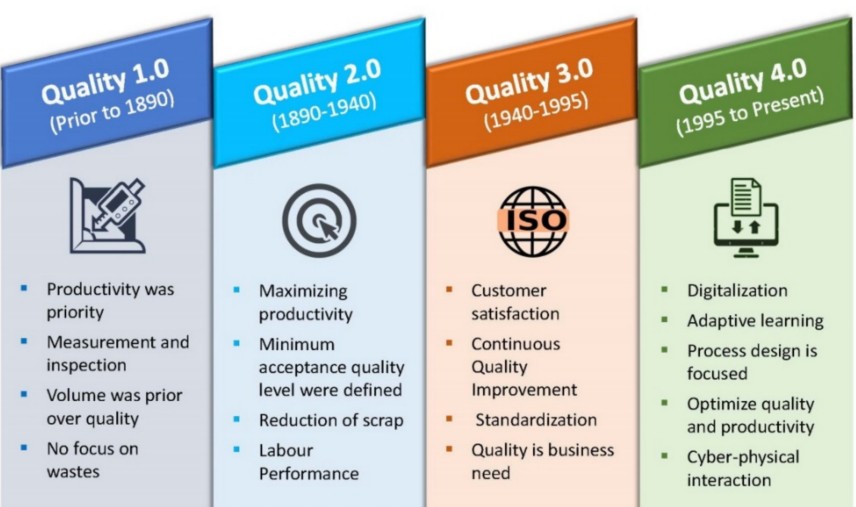

**Figure 1.** Digital transformation from Quality 1.0 to Quality 4.0.

In the beginning of the industrialization era, Quality 1.0, quality was based primarily on inspections and measurements, and the primary focus was on productivity with higher volumes. In the second era, named Quality 2.0, maximizing labour productivity was the priority, but the minimum acceptance levels for quality were defined, and focus shifted towards reducing scrap and waste. In the era of Quality 3.0, customer satisfaction and standardization were fully adopted. Concepts of continuous quality improvements and Deming's PDSA cycle were adopted, and it was realized that quality is necessary for every business to generate customer satisfaction and hence business growth. In the current era of Quality 4.0, digitalization and the adoption of smart techniques are the focus for developing autonomous systems for the optimal trade-off between quality and productivity [11].



Industries have an immense need for the state-of-the-art tools and technologies for agile responses to the shifting field of quality management. In this regard, a comprehensive literature review has been conducted to evaluate the advancements and gaps in industry for the implementation of the latest requirements of quality management systems.

Extensive debate led to implementing the total quality management (TQM) philosophy throughout organizations to produce benefits such as reduced waste, increased revenues, and strategic advancements. Researchers mainly focused on the standardization or reengineering of processes and organizations' strategic alignment with customer satisfaction. The core outcome of the research encompassed a framework that was validated through implementations in the manufacturing and service industries [12–18]. Industries had a critical need to adopt digitalization and real-time data management systems to embed within enterprise-wide resource planning. The Boston Consulting Group (BCG) presented a comprehensive report encompassing the current conditions and a SWOT analysis of the global industry in context of Quality 4.0 [19]. Based on that report, the framework for Quality 4.0 was developed by researchers, professionals, and other stakeholders who outlined the components, tools, and techniques. The ASQ addressed the evolution of Quality 4.0 and are the main enablers of its implementation in industry [11]. A blog by J. M. Juran presented a comprehensive framework for the implementation of Quality 4.0, defining the eleven axes, and their tools and techniques, that are necessary for achieving excellence in Quality 4.0 [20,21]. Gohane et al. and Theuws reviewed a comprehensive body of literature and presented the concept and transformation of Quality 4.0. They discussed the issues and requirements related to the implementation of Quality 4.0 [22,23]. Sarder et al. presented research outlining the qualitative and quantitative measures through which the impacts of digitalization on quality performance in organizations can be determined [24]. Zonnenshain and Kenneth emphasized that Quality 4.0 is mainly a data-driven discipline and that there is a need for enterprise-wide data sciences implementation to obtain real-time data from all business transactions [25]. Escobar et al. presented research highlighting the contributions of big data and artificial intelligence (AI) in the field of Quality 4.0. They found that about 80 to 87 percent of big data projects could not deliver value and that most of industry leaders lack the vision to utilize the big data and AI technologies [26]. Corti et al. addressed the core issues in implementing Quality 4.0 in the era of Industry 4.0. They also proposed a framework for the implementation of Quality 4.0 in industry and validated the same through the outcomes of implementation in a model organization [27]. Chiarini et al. in an exploratory study performed questionnaire-based research on two different organizations and assessed their capabilities according to the Quality 4.0. They concluded that the implementation of Quality 4.0 is totally dependent upon the skills of employees and quality managers. The main components of Q4.0 are people, processes, and technology, and people play the dominant role in positive outcomes of Quality 4.0 [28]. Santos et al. proposed that human resources skills are required for the transformation towards Quality 4.0. Using survey-based research in different countries, they formulated the mandatory skill set for human resources: an understanding of information and communication technology (ICT), big data analysis, team and individual management, and common traits of quality management [29]. Ali and Johl conducted a comprehensive literature review and presented the required factors for the implementation of Industry 4.0 for total quality management. The main factors are the commitment of leadership, customer satisfaction, organizational learning culture, and big data analytics [30,31]. Florencio de Souza et al. coined the term Total Quality Management 4.0, following a literature review, referring to processes that are quite similar to Quality 4.0 but that differ in implementation [32]. Glogovac et al. developed a maturity measurement model for the quality in Industry 4.0. The model is mainly based on the compliance of the system with ISO 9004: 2018. The model presents a holistic view of quality management [33]. For the sustainability of Industry 4.0 and Quality 4.0 implementation, several researchers presented their frameworks for knowledge-intensive business processes (KIBPs). The frameworks primarily discuss KIBPs in the context of knowledge management and analytics. The effective implementation of

KIBPs may result in productive outcomes [34–36]. Researchers have also addressed the legal and environmental issues that can arise in implementing Industry 4.0 and Quality 4.0. Previous studies correlated allied technologies (pillars) of Industry 4.0 with negative impacts such as air pollution and harmful waste discharge that are noncompliant with UN Sustainable Development Goals (SDGs) [37]. Another study provides the foundation for improvements in corporate sustainability through environmental aspects. Research presented the influence of environmental sustainability on Industry 4.0, resulting more accurate, real-time environmental management in developing industry visions [38]. Based on the above-mentioned literature review, the concept of Quality 4.0 is not new, but its implementation with all its components is quite cumbersome. The literature presents the framework for the implementation of Quality 4.0 and its components but does not outline the actions required for implementation. The current literature presents the strengths and weaknesses of industry implementations of Quality 4.0 in developed environments. Hence, the core idea of this research is to outline the gaps in the implementation of Quality 4.0 in developing countries such as Pakistan with respect to the following research questions:

- How can industries implement Quality 4.0 in developing countries?
- What is the extent of industry implementation of the overall Quality 4.0 framework?
- What are the factors influencing the implementation of Quality 4.0 in industry?

The methodology of the research starts with the literature review and the identification of the research problem, based on which the research questions and the hypothesis are defined. A research instrument was developed to gauge the strengths, weaknesses, opportunities, and threats related to industry Quality 4.0 implementation in developing countries. The research instrument is a questionnaire which measures the components of Quality 4.0 in terms of knowledge, understanding, application, and effectiveness. The content validity index (CVI) of the research questionnaire was measured to determine the relevance of the questions for the research objectives and the ease of understanding of the questions. With a pilot survey, Cronbach's alpha was also determined to evaluate the internal consistency of the responses. The respondents' answers were statistically analysed to check the significance of each variable and validity of the hypothesis. Finally, the results are presented in a SWOT matrix outlining the strengths, weaknesses, opportunities, and threats related to industry implementation of Quality 4.0. An illustration of the methodology is presented in Figure 2.

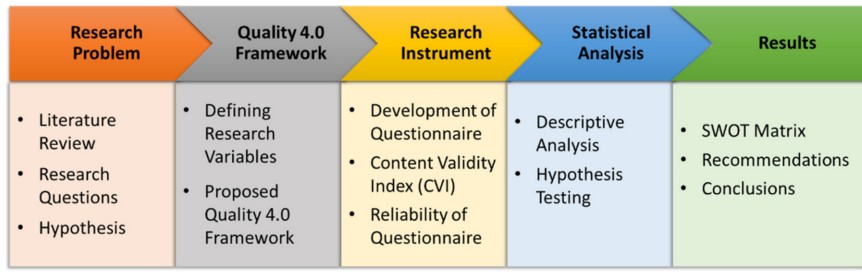

**Figure 2.** Proposed research methodology.

To address the research questions, a set of hypotheses was proposed as follows:

**Hypothesis 1 (H1).** *Quality management practitioners confirm that the Quality 4.0 framework is being implemented in industry.*

**Hypothesis 2 (H2).** *Quality management practitioners perceive that all eleven dimensions of the Quality 4.0 framework are being implemented in industry.*

**Hypothesis 3 (H3).** *Quality management practitioners perceive that the eleven exes of Quality 4.0 have relationships among each other.*

**Hypothesis 4 (H4).** *Quality management practitioners perceive that the Quality 4.0 framework is well implemented in multinational organizations.*

**Hypothesis 5 (H5).** *Quality management practitioners perceive that the Quality 4.0 framework is well implemented in large organizations having more than 500 employees.*

**Hypothesis 6 (H6).** *Quality management practitioners perceive that the presence of quality management professionals enables an organization to implement the Quality 4.0 framework.*

In the current article, five major sections are identified. The Section 1 discusses the introduction, literature review, and research questions. The Section 2 describes the developed framework in detail. The Section 3 presents the variables to be measured for the assessment of framework implementation with the research instrument developed. The validity and reliability of the research instrument are also discussed in detail. The Section 4 presents and discusses the results of the statistical analysis of the research instrument response data. The Section 5 presents the strengths, weaknesses, opportunities, and threats related to Quality 4.0 industry implementation in developing countries along with recommendations for improvements. The last section summarizes the research and presents the conclusion of the work.

## 2. Conceptualization and Implementation of Quality 4.0 Framework

The core objective of this work is to develop a Quality 4.0 framework that will enable shifting from traditional to advanced quality assurance systems. The framework is based on the eleven dimensions of the Quality 4.0 as presented by Juran [20]:

| | | | |
|---|---|---|---|
| (i) Compliance | (iv) Connectivity | (vii) Management System | (x) Data |
| (ii) Competency | (v) Collaboration | (viii) Web-based Applications | (xi) Analytics |
| (iii) Leadership | (vi) Culture | (ix) Scalability | |

Each element of the Quality 4.0 framework is applied through the tools and techniques as defined in the presented literature [20,21]. The framework is not limited to the manufacturing industry but can apply equally in the service industry. The Quality 4.0 framework proposed in this research outlines all eleven dimensions, presented in Figure 3 and discussed in detail.

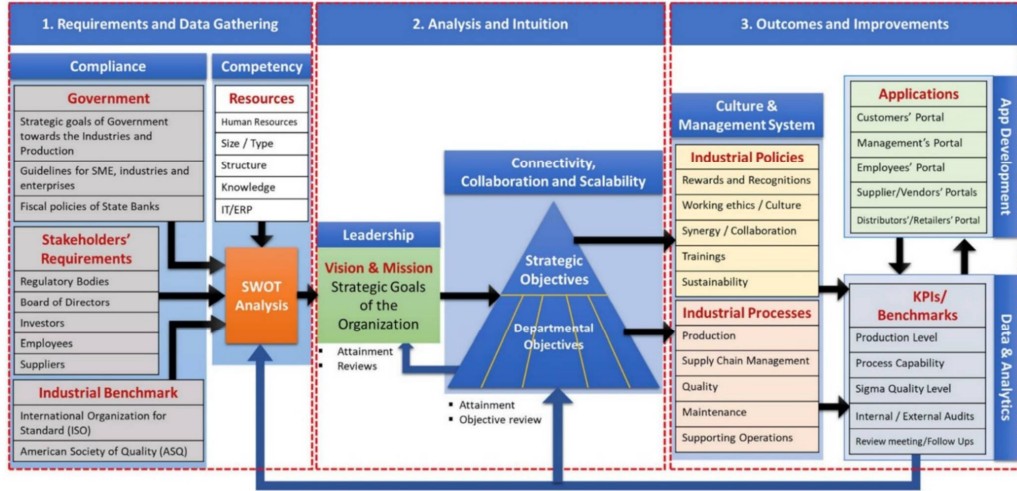

**Figure 3.** Proposed Quality 4.0 framework.

### 2.1. Compliance

Conventionally, SWOT analysis is performed by gathering the needs and requirements of stakeholders, (either external or internal) to analyse the current strengths and weaknesses of the organization and the opportunities and threats in the industry, which can affect

the objective of any organization. The opportunities and threats are analysed based on environmental scanning (external factors).

In the Quality 4.0 framework, SWOT analysis can be performed using artificial-intelligence-based (AI-based) bots. This analysis encompasses the framework dimensions of compliance and competency. Compliance refers meeting government obligations and other standards, stakeholder requirements, and industry benchmarks [20]. The government obligations are mainly related to individual ministries of industries, which plan and assign strategic goals to different industries and set targets for exports, production level, energy consumption, labour market factors, and other parameters. Guidelines and annual plans for SMEs and other enterprises are disseminated by regulatory bodies. The fiscal policies of state or reserve banks define interest rates and export targets, which are the major inputs to the strategic planning of any organization. Regulatory bodies such as state banks, federal board of tax and revenues, and labour and human resource agencies outline the organizational constraints related to loads, taxes, labour laws, and other factors. Boards of directors, employees, trade unions, suppliers, and investors are the core stakeholders and are focused on productivity, revenues, and constraints in strategic plans. Industry benchmarks are based on best practices and thus usually serve as targets for organizations, for instance to reduce defects, increase the value to the customer, improve process capabilities, or other goals.

AI-based bots are pre-programmed computer programmes that are fed relevant keywords, such as industry-specific terms, competitors' names, rule and law terminology, and minutes of meetings of relevant bodies. The bots search the information from the websites of relevant entities such as ministries of production, law, labour, environment, and finances; industry associations, the International Organization for Standardization (ISO), the American Society for Quality (ASQ), and others. The quantitative and qualitative data are acquired, filtered, and analysed by AI algorithms under the supervision of personnel who tweak and provide the essential information to the bots, which perform the environmental scanning around any organization. The personnel and bots in the Quality 4.0 framework are called business intelligence units (BIUs).

### 2.2. Competency

Another input to the SWOT analysis is the internal factors of the organization, its strengths and weaknesses. These factors can be analysed using the AI-based bots, which acquire the data from enterprise-resource planning (ERP) modules. These internal factors include the knowledge and skills of the employees and the organization's structure, type, size, and location. Supply chains, quality management systems, customer complaints and their resolution, and information technology implementation play important roles in determining the strengths and weaknesses in any organization. The BIUs compare actual performance with defined benchmarks and continuously update practices based on the key performance indicators (KPIs) through the modules of the ERP. The deviation between the benchmarks and the KPIs defines the strength and weaknesses of the organizations, and live dashboards present the overall SWOT analysis findings.

The SWOT analysis is performed based on the internal and external factors (qualitative data) highlighted by the BIUs. The factors are assigned weights based on their importance as highlighted by the industry through the AHP or other techniques, and scores are assigned by the BIU following the pre-defined rubrics. Organizations that comply with the findings from the real-time data acquisition and SWOT analysis are strong on the dimensions of compliance and competency in the Quality 4.0 framework.

### 2.3. Leadership

The next stage of the Quality 4.0 framework is to set the vision and mission of the organization based on the SWOT analysis findings of the compliance and competency of the organization. The process involves the organizational knowledge base and the intuition and far-sightedness. Based on the SWOT analysis and the data from the BIU, forecasts of the

organization's revenue, product/service demand, resources, and strategy can be proposed. Using the gathered qualitative and quantitative data, and business intelligence modules, the forecasts can determine the organization's strategies. The knowledge, involvement, and control of leadership in strategic planning and operations determines an organization's strength on this dimension of the Quality 4.0 framework.

### 2.4. Connectivity

The connectivity in the proposed Quality 4.0 framework is the existence of quality circles in the organization comprising representatives from different departments. These representatives may be connected through an ERP system, where data related to quality are input by quality and other departments, and reports are generated using quality tools. Quality circle meetings are held frequently, either physically or remotely, to monitor and control the quality of product and services. Real-time quality reports are presented via ERP systems using quality tools such as histograms, pareto charts, control charts, and FMEA. The connectivity dimension of the Quality 4.0 framework ensures that all department in an organization are working in the same strategic direction as decided by the management.

### 2.5. Collaboration

The strategic plan of an organization is decided by the leadership based on the vision and mission of the organization. Hence, it is quite necessary to work together in the same direction to attain the long-term objectives of the organization. In the proposed Quality 4.0 framework, the strategic objectives developed by the leadership are communicated to the organization through digital dashboards, signs, and screens in the different departments. These could display the organization's vision, mission, slogans, teamwork, targets, rewards, and individual achievements to enhance productivity and collaboration. Board of directors meetings provide a forum for the leadership across the whole organization to come together to discuss the quality performance of the organization. The frequency of these review meetings, the agenda items, the follow-up actions, and the measurements against the performance indicators can indicate the strength of the organization on the collaboration dimension of the Quality 4.0 framework.

### 2.6. Scalability

Scalability is another dimension in the Quality 4.0 framework; it assesses the effectiveness of implementing the framework irrespective of the size of the organization. It encompasses the size of data generated through the organization and capabilities to analyse the data using cloud systems. In the Quality 4.0 framework, departments and divisions measure their achievement of the KPIs and departmental objectives. Scalability assesses the level, size, velocity and type of data which is being gathered to generate the KPIs. It may include the data from the industrial internet of things (IIoTs) and the other IoTs within the organization.

### 2.7. Culture

Culture in the proposed Quality 4.0 framework refers to the organization's culture of quality. This dimension of the Quality 4.0 framework can be assessed through the existence of quality circles and the use of quality management tools, as well as the knowledge and understanding to act on the inferences from the outcomes of these tools. It includes the confidence of personnel in the organizational processes and procedures and the confidence of the organization in its personnel. Culture includes offering rewards and recogfor performance improvements.

### 2.8. Management System

The quality management system is the conventional part of the Quality 4.0 framework and covers the existence and implementation of standardization, in the form of SOPs, quality audits, improvements, the involvement of leadership, corrective and follow-up

actions, etc. This management system in the Quality 4.0 framework relies on technologies that allow leadership to observe the real-time statistics, dashboards, and KPIs of the organization. The communication of organizational policies, SOPs, and KPIs may be easier using ERP-based systems in the Quality 4.0 framework.

### 2.9. App Development

This is one of the most important dimensions of the Quality 4.0 framework; it integrates the whole organization through portals and applications installed on employees' PCs and mobile phones. Real-time data related to the quality such as number of units produced, types of defects, rework, calibrations, and violated SOPs can be communicated for analysis. Data from customers, retailers, and distributors such as complaints can be acquired and analysed. The same may be communicated to suppliers and vendors in real time through the portals and applications. Through this app development, leadership can view the real-time quality statistics and reports of the organization.

### 2.10. Data Gathering

This is the most important dimension of the Quality 4.0 framework. Data can be gathered through the IIoTs and other IoTs in the organization for analysis related to KPIs. These data can be acquired from sensors installed to detect numbers and types of defects, reworks, calibrations, setup and idle times, productivity, etc. The data are filtered using the AI-based programmes, which eliminates the non-credible data and keeps the relevant data. The data are analysed through the programmes, which are developed using machine learning techniques and provide the process capabilities in real time. Data gathering is the main aspect of the whole framework because based on the analysis of the data, the objectives, strategies, and responses to the SWOT analysis can be improved.

### 2.11. Analytics

Big data analytics is one the major aspects of Quality 4.0; the data from IIoTs and other sources are uploaded to cloud storage and analysed using artificial intelligence-based algorithms to recommend or make the decisions. The quality data relate mainly to rejections and numbers, types, and causes of defects. Inferences are made from the quality reports produced based on the analysis of the data that include recommendations.

In the following section, Pakistani industry was assessed for the maturity of its implementations of the Quality 4.0 framework.

### 3. Research Instrument and Sampling

To determine the maturity of the industry based on the Quality 4.0 framework, a survey research instrument was developed to measure the key research variables according to the framework, presented in Table 1.

**Table 1.** Research variables mapped to the dimensions of the Quality 4.0 framework.

| S. No. | Dimension | Survey Question | Variables |
|---|---|---|---|
| 1. | | Organization Name (Open response) | ORGAN |
| 2. | | Type of Organization (Automotive, Textile, IT, Health Care, Food, Energy, other) | ORGTYP |
| 3. | | Organization Category (Government, Semi-Government, Private) | ORGCAT |
| 4. | – | Designation (Supervisor, Asst. Manager, Manager, Sr. Manager, Gen. Manager, Director, Other) | DESIGN |
| 5. | | Department Name (Supply Chain, Maintenance, Quality, Production, Maintenance, Other) | DEPTT |
| 6. | | No. of Employees (Less than 100, 101 to 500, more than 500) | EMPL |
| 7. | | Organization Scope (Domestic, Multinational) | ORGSC |
| 8. | | No. of Quality Professionals having professional certification in the field of quality management | NOQP |

**Table 1.** *Cont.*

| S. No. | Dimension | Survey Question | Variables |
|---|---|---|---|
| 9. | | Strategic goal of the government (Full—5, Major—4, Normal—3, Minor—2, None—1) | CMPL1 |
| 10. | | Guidelines of SME, Industries and Enterprises | CMPL2 |
| 11. | | Fiscal policies of the state/reserve banks | CMPL3 |
| 12. | Compliance | Regulatory Bodies | CMPL4 |
| 13. | (CMPL) | Decisions of Board of Directors | CMPL5 |
| 14. | | Decision of Investors | CMPL6 |
| 15. | | Requirements of Employers | CMPL7 |
| 16. | | Requirements of Suppliers | CMPL8 |
| 17. | | Implementation of Quality Standards | CMPL9 |
| 18. | | Knowledge of Quality Assurance (Highest—5, Major—4, Average—3, Normal—2, None—1) | COMT1 |
| 19. | Competency | Competency of Human Resource related to Technology | CMPT2 |
| 20. | (CMPT) | Formalization of Organizational Structure | CMPT3 |
| 21. | | Organizational Knowledge | CMPT4 |
| 22. | | Smart Phones/Gadgets, IT Infrastructure and ERP | CMPT5 |
| 23. | | Connectivity through the IT Infrastructure (Highest—5, Major—4, Normal—3, Minor—2, None—1) | CONN1 |
| 24. | | Connectivity of Strategic and Departmental Objectives | CONN2 |
| 25. | Connectivity | Connectivity through ERP and its linkage with the Quality | CONN3 |
| 26. | (CONN) | Connectivity of Strategic and Departmental Data to form KPIs | CONN4 |
| 27. | | Connectivity of Smart Phones/Gadgets with the QMS | CONN5 |
| 28. | | Connectivity of Quality System with the IT Infrastructure | CONN6 |
| 29. | | Existence of Quality Circle and its meeting (Highest—5, Major—4, Normal—3, Minor—2, None—1) | COLL1 |
| 30. | Collaboration | KPIs/Quality Report sharing on regular basis | COLL2 |
| 31. | (COLL) | Existence and effectiveness of Quality Board of Review | COLL3 |
| 32. | | Shared responsibilities of follow-ups related to Quality | COLL4 |
| 33. | | Ease in processing small to large-scale data (Highest—5, High—4, Normal—3, Low—2, None—1) | SCAL1 |
| 34. | Scalability | Capability to process small to large-scale manufacturing | SCAL2 |
| 35. | (SCAL) | Capability of firm to acquire product/variant wise quality data | SCAL3 |
| 36. | | Involvement in SWOT Analysis (Full-time—5, Mostly—4, Normal—3, A few—2, None—1) | LEAD1 |
| 37. | Leadership | Leadership alignment with the Vision and Mission | LEAD2 |
| 38. | (LEAD) | Engagement with the Quality Management Systems | LEAD3 |
| 39. | | Seriousness towards the Quality 4.0 framework | LEAD4 |
| 40. | | Working culture towards the quality (Highest—5, High—4, Normal—3, Low—2, None—1) | CULT1 |
| 41. | Culture | Reward and recognition for improvements in quality | CULT2 |
| 42. | (CULT) | Culture of synergy, collaboration and flexibility | CULT3 |
| 43. | Management | Existence of QMS (Highest—5, High—4, Normal—3, Low—2, None—1) | MGS1 |
| 44. | System | Frequent Trainings of Quality Management System (QMS) | MGS2 |
| 45. | (MGS) | Review Meetings for the performance assessment QMS | MGS3 |
| 46. | | Internal and External Audits for the performance reviews of QMS | MGS4 |
| 47. | App | Existence of portals for data collections and reporting | WAPP1 |
| 48. | Development | Ease of using portals (Complete—5, Major—4, Normal—3, Minor—2, None—1) | WAPP2 |
| 49. | (WAPP) | Frequency of portal updates | WAPP3 |
| 50. | Data | Real-time collections of data from different departments | DATA1 |
| 51. | (DATA) | Size of data (Highest—5, High—4, Average—3, Low—2, Least—1) | DATA2 |
| 52. | | Reliability of data | DATA3 |
| 53. | | Availability of cloud-based data analytics (Highest—5, High—4, Average—3, Low—2, Least—1) | ANLY1 |
| 54. | Analytics | Use of machine learning or artificial intelligence-based analysis | ANLY2 |
| 55. | (ANLY) | Existence of real-time dashboards accessible to the leadership and management | ANLY3 |
| 56. | | Flexibility in analysis of data | ANLY4 |

Initially, 73 questions were developed, and the final survey comprised 56 survey items. Each item, all mapped to a variable, were rated on Likert scales from 1 to 5. Table 2 presents the explanations of the variable responses for the better understanding of the respondents.

**Table 2.** Rubrics for the research variables.

| Dimension | Response Option | Description |
|---|---|---|
| Compliance (CMPL) | Full—5 | Full compliance with the assessments from the internal and external audits |
| | Major—4 | Major compliance with the assessments from the audits |
| | Normal—3 | Normal compliance with the assessments |
| | Minor—2 | Minor compliance with assessments |
| | None—1 | No compliance |
| Competency (CMPT) | Highest—5 | Highest knowledge, with professional certification or benchmark standard |
| | Major—4 | Major knowledge, with formal certification or benchmark standard |
| | Average—3 | Average knowledge, with self-learning and informal skills |
| | Minor—2 | Normal knowledge, with self-learning |
| | None—1 | No knowledge or skills |
| Connectivity (CONN) | Highest—5 | Highest connectivity, with real-time data gathering and reporting |
| | Major—4 | Major connectivity, with data gathering and reporting updates within 1 to 2 h |
| | Normal—3 | Normal connectivity, with daily data gathering and reporting |
| | Minor—2 | Static connectivity, with weekly data gathering and reporting |
| | None—1 | No connectivity at all |
| Collaboration (COLL) | Highest—5 | Continuous communication online or via daily meetings |
| | Major—4 | Major communication through informal social media |
| | Normal—3 | Normal communication over phones or VOIP |
| | Minor—2 | Low communication through weekly meetings |
| | None—1 | Very little communication |
| Scalability (SCAL) | Highest—5 | Very quickly with a click |
| | High—4 | Some work is needed to transform data from small to large scale |
| | Normal—3 | An IT professional is needed to transform data from small to large scale |
| | Low—2 | The two types of data are handled separately |
| | None—1 | Only one type of data is handled |
| Leadership (LEAD) | Full-time—5 | Leadership is totally engaged with the management or real-time actions are taken |
| | Mostly—4 | Leadership has several businesses and partially engaged with the management |
| | Normal—3 | Leadership looks after the business and engages with the management on weekly basis |
| | A few—2 | Leadership seldom looks after the business or once a month |
| | None—1 | There is no engagement of leadership at all |
| Culture (CULT) | Highest—5 | Leadership and employees practice the culture habitually |
| | High—4 | The culture is practiced informally in most of the divisions |
| | Normal—3 | The culture is practiced occasionally and only with leadership will |
| | Low—2 | A few managers at their own level practice the culture |
| | None—1 | No such culture exists |
| Management System (MGS) | Highest—5 | A formal system exists and the leadership is committed to practicing the same |
| | High—4 | A formal system exists but the leadership is partially committed to practicing the same |
| | Normal—3 | A formal system exists but no leadership commitment exists |
| | Low—2 | There is an informal system |
| | None—1 | There is no system at all |
| App Development (WAPP) | Complete—5 | A paperless system exists in the organization with ease and real-time connectivity |
| | Major—4 | A portal exists but the data are entered manually |
| | Normal—3 | A few islands of portal exist which are not well connected |
| | Minor—2 | Major tasks are performed on the apps but others are performed manually |
| | None—1 | There is no app development |

**Table 2.** *Cont.*

| Dimension | Response Option | Description |
|---|---|---|
| Data (DATA) | Highest—5 | Instantaneous data gathering or data size > 1 TB or no filtering needed |
| | High—4 | Data are updated every hour or data size is greater than 1 GB and less than 1 TB or little filtering is required |
| | Normal—3 | Data are updated every day or data size is less than 1 GB or filtering takes a full day |
| | Low—2 | Data are updated every week or data size is less than 1 GB or filtering takes several days |
| | None—1 | Data are static and is on papers or data size is less than 1 GB or selective data is taken |
| Analytics (ANLY) | Highest—5 | Exists and is fully implemented in the organization |
| | High—4 | Exists and implemented for critical data |
| | Normal—3 | Exists and implemented for quarterly or bi-annual data |
| | Low—2 | Exists and implemented for annual data |
| | None—1 | Does not exist |

The survey questionnaire was reviewed by three experts in the field to produce a content validity index (CVI). Out of 73 questions, the experts agreed on 56 questions, and the universal agreement CVI (S-CVI/UA) was calculated as 0.711, which is higher than the benchmark, 0.7. Hence, the questionnaire was found to be appropriate for conducting the research.

The reliability of the responses was measured with a sample survey of 20%. The Cronbach's alpha for the sample survey was 0.785. Since the value was higher than the benchmark of 0.7, the questionnaire was reliable and could be used for further analysis.

The convenience sampling technique was adopted in which the respondents targeted were leaders and managers, who are significant responsible for quality management system. Data were collected from small, medium, and large organizations. A sample size of 380 participants was targeted for assessing the maturity of the implementation of Quality 4.0 in third world countries such as Pakistan. A total of 213 responses were submitted, and 161 responses were found complete and appropriate to be analysed. The distribution of the respondents according to their industry type, category, designation and number of employees is presented in Figure 4.

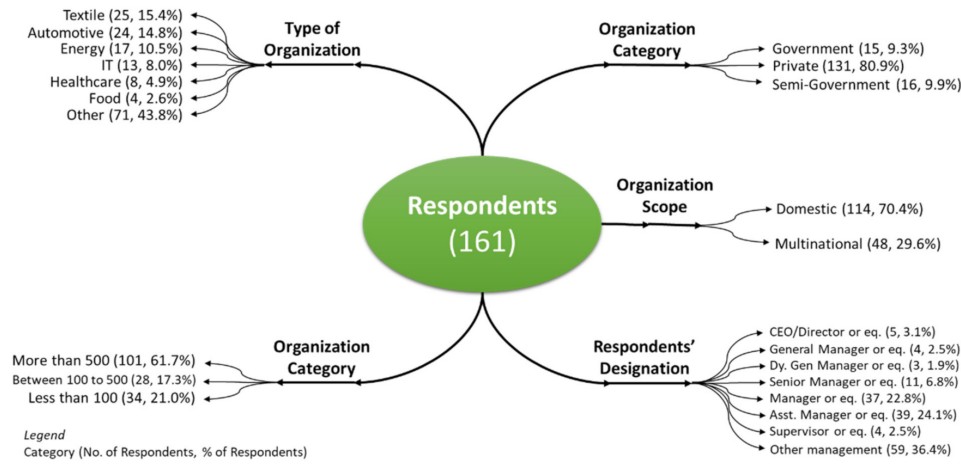

**Figure 4.** Number and percentage of respondents according to their organization's industry, category, and scope and their designation.

## 4. Statistical Analysis & Results

The 161 complete responses to the 56 survey questions were statistically analysed to determine the level of implementation of the Quality 4.0 framework in the different industries.

For each dimension of the Quality 4.0 framework, the average scores were computed for each item to determine the status of each organization in implementing the Quality 4.0 framework. As shown in Table 2, the response option 2 reflects minor implementation of the Quality 4.0 dimension, and 3 indicates the normal implementation of the dimension. Hence, the average of these two scores is 2.5, which can be taken as minor to normal implementation of the Quality 4.0 dimension. Figure 5 shows the average scores for each dimension for the different industries.

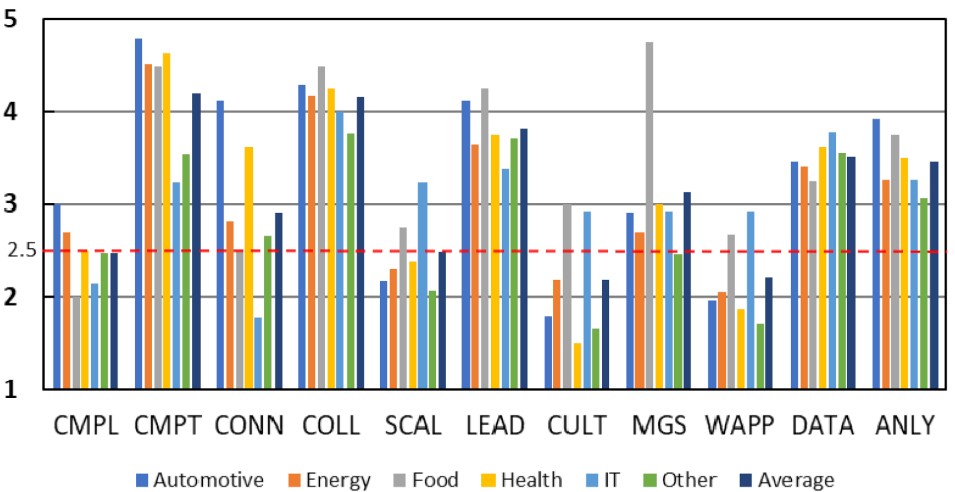

**Figure 5.** Mean scores for each of the eleven Quality 4.0 dimensions by type of organization.

The inferential summary of Figure 5 is presented in Table 3.

**Table 3.** Implementation of the Quality 4.0 Dimensions According to the Type of Organization.

| Industry Type | Dimension of Quality 4.0 | | |
| --- | --- | --- | --- |
| | **Major Implementation (Average Score 4 to 5)** | **Average Implementation (Average Score 2.5 to 3.9)** | **Minor Implementation (Average Score 1 to 2.4)** |
| Automotive | Competency, Connectivity, Collaboration, and Leadership | Compliance, Data Gathering, Analytics, and Management System | Scalability, Culture, and App Development |
| Energy | Competency and Collaboration | Compliance, Connectivity, Leadership, Management System, Data Gathering, and Analytics | Scalability, Culture, and App Development |
| Food | Competency, Collaboration, Leadership, and Management System | Connectivity, Scalability, Culture, App Development, and Data Analytics | Compliance |
| Health | Competency and Collaboration | Connectivity, Leadership, Management System, Data Gathering, and Analytics | Compliance, Scalability, Culture, and App Development |
| IT | None | Competency, Collaboration, Scalability, Leadership, Culture, Management System, App Development, Data Gathering, and Analytics | Compliance and Connectivity |
| Others | None | Competency, Connectivity, Collaboration, Leadership, Data Gathering, and Analytics | Compliance, Scalability, Culture, Management System, and App Development |

Figure 6 shows the average scores for each dimension according to the industry category.

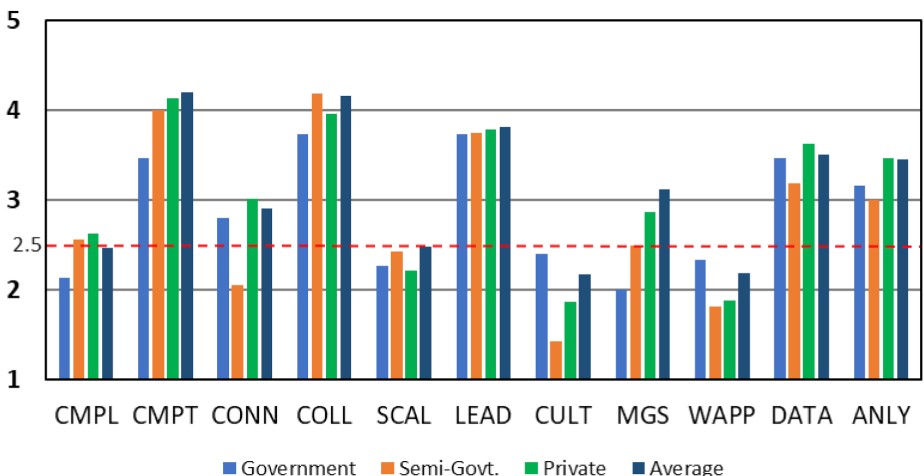

**Figure 6.** Mean scores for each of the eleven Quality 4.0 dimensions by organization category.

The inferential summary of Figure 6 is presented in Table 4.

**Table 4.** Implementation of the Quality 4.0 Dimensions According to the Category of Organization.

| Industry Category | Dimension of Quality 4.0 | | |
|---|---|---|---|
| | **Major Implementation (Average Score 4 to 5)** | **Average Implementation (Average Score 2.5 to 3.9)** | **Minor Implementation (Average Score 1 to 2.4)** |
| Government | None | Competency, Connectivity, Collaboration, Leadership, Data Gathering, and Analytics | Compliance, Scalability, Culture, Management System, and App Development |
| Semi-Government | Competency and Collaboration | Compliance, Leadership, Management System, and Data Analytics | Connectivity, Scalability, Culture, and App Development |
| Private | Competency | Compliance, Connectivity, Collaboration, Leadership, Management System, Data Gathering, and Analytics | Scalability, App Development, and Culture |

Figure 7 shows the average scores for each dimension according to the organization's number of employees.

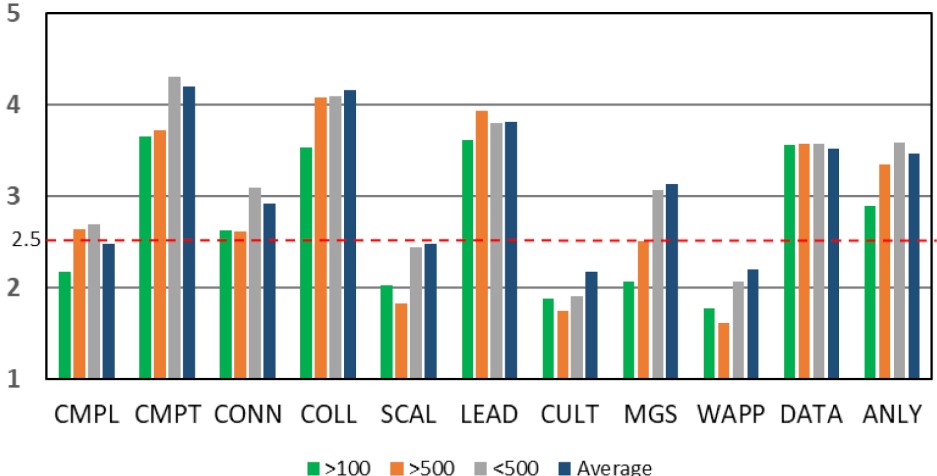

**Figure 7.** Mean scores for each of the eleven Quality 4.0 dimensions by number of employees.

The inferential summary of Figure 7 is presented in Table 5.

**Table 5.** Implementation of the Quality 4.0 Dimensions According to the Size of Organization.

| No. of Employees | Dimensions of Quality 4.0 | | |
| --- | --- | --- | --- |
| | **Major Implementation (Average Score 4 to 5)** | **Average Implementation (Average Score 2.5 to 3.9)** | **Minor Implementation (Average Score 1 to 2.4)** |
| Less than 100 | None | Competency, Connectivity, Collaboration, Leadership, Data, Analytics | Compliance, Scalability, Culture, Management System, and App Development |
| Between 100 to 500 | Collaboration | Compliance, Competency, Connectivity, Leadership, Management System, Data Gathering, and Analytics | Scalability, Culture, and App Development |
| More than 500 | Competency and Collaboration | Compliance, Connectivity, Leadership, Management System, Data Gathering, and Analytics | Scalability, Culture, and App Development |

Figure 8 shows the average scores for each dimension according to the organization's scope.

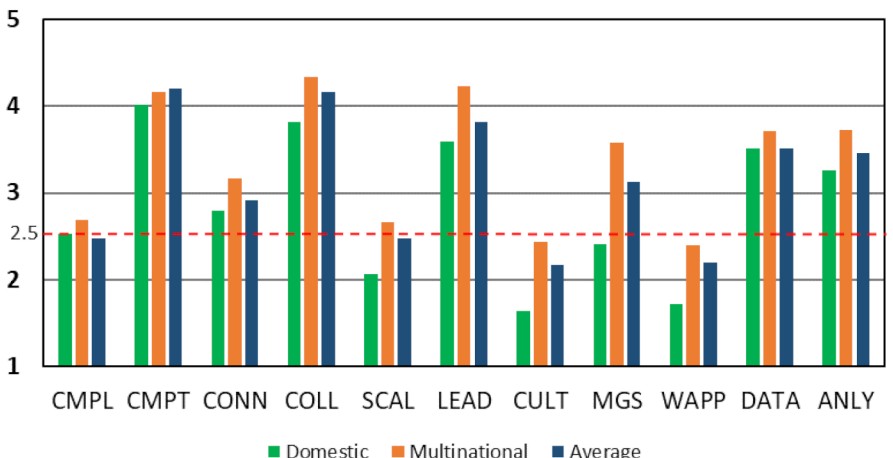

**Figure 8.** Mean scores for each of the eleven Quality 4.0 dimensions by organization scope.

The inferential summary of Figure 8 is presented in Table 6.

**Table 6.** Implementation of the Quality 4.0 Dimensions According to the Scope of Organization.

| Scope of Organization | Dimension of Quality 4.0 | | |
| --- | --- | --- | --- |
| | **Major Implementation (Average Score 4 to 5)** | **Average Implementation (Average Score 2.5 to 3.9)** | **Minor Implementation (Average Score 1 to 2.4)** |
| Domestic | Competency | Compliance, Connectivity, Collaboration, Leadership, Data Gathering, and Analytics | Scalability, Culture, Management System, and App Development |
| Multinational | Competency, Collaboration, and Leadership | Compliance, Connectivity, Scalability, Management System, Data Gathering, and Analytics | Culture and App Development |

The overall mean scores for the eleven dimensions of the Quality 4.0 framework implementation are presented in Figure 9.

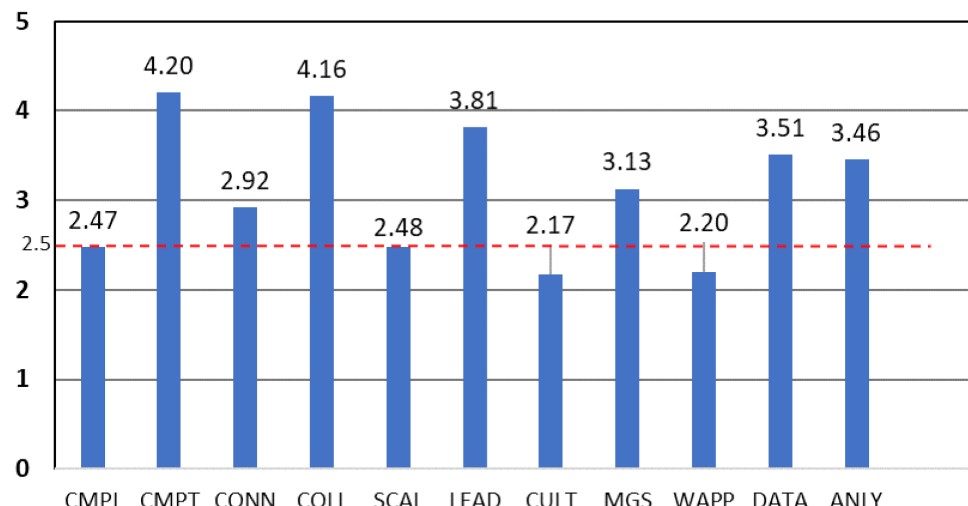

**Figure 9.** Overall mean scores for the eleven Quality 4.0 dimensions.

From Figure 9, the overall mean (mean of all the dimensions by the respondents) shows a value of 3.01, which lies between the minor to normal implementation of the eleven dimensions, which supports H1. That is, the implementation of Quality 4.0 framework is significant in the industry. However, to assess the trueness of other hypothesis, the mean responses of each eleven dimensions of the Quality 4.0 framework are discussed.

The mean responses (see Figure 9) shows that the highest industry maturity (mean score (MS) between 4 and 5) is in competency (MS = 4.06) and collaboration (MS = 3.97), which is also evident from Figure 5 to Figure 8, especially in multinational and private firms, whereas it is lacking in the government sector industries (see Figure 5). The average normal implementation scores (between 2.5 to 3.99) were for the Quality 4.0 dimensions of compliance (MS = 2.57), connectivity (MS = 2.91), leadership (MS = 3.78), management system (MS = 2.76), data gathering (MS = 3.57), and analytics (MS = 3.40), which is also evident from Figure 5 to Figure 8. The lowest mean scores for the Quality 4.0 dimensions were for scalability (MS = 2.25), culture (MS = 1.88), and app development (MS = 1.92). It is also evident from Figure 5 to Figure 9 that according to the industry type, category, size and respondent scope, these three dimensions lack implementation in Pakistani industry. Compliance was the major issue across industry types and categories, even food (MS = 2.00, Figure 5), which is the most sensitive industry for compliance with standards. These findings clearly reject H2. Industries need to work on the implementation of the Quality 4.0 dimensions, specifically those with the lowest mean scores (MS), to meet the current requirements of global industry.

To validate the findings for H1 and H2 related to the eleven dimensions of Quality 4.0, and a target value of 2.5 (minor to normal implementation) was set to compare the means of the responses for each variable. Table 7 provides the mean, standard deviation, t-value, significance (*p*-value), and mean difference for each dimension. In this table, if the significance (*p*-value) of any dimension is less than 0.05, it shows that the response is within the 95% confidence interval and is significantly higher than the set value of 2.5 (minor to normal implementation).

**Table 7.** *t*-test Statistics for the Eleven Dimensions of Quality 4.0.

| Dimension | Mean | St. Dev. | SE of Mean | DOF | t-Value | Significance (*p*-Value) |
|---|---|---|---|---|---|---|
| CMPL | 2.574 | 1.091 | 0.086 | 161 | 0.864 | 0.806 |
| CMPT | 4.062 | 1.350 | 0.106 | 161 | 14.725 | 1.000 |
| CONN | 2.907 | 1.360 | 0.107 | 161 | 3.814 | 1.000 |
| COLL | 3.969 | 1.111 | 0.087 | 161 | 16.826 | 1.000 |
| SCAL | 2.247 | 1.462 | 0.115 | 161 | −2.203 | 0.014 |
| LEAD | 3.784 | 1.265 | 0.099 | 161 | 12.924 | 1.000 |
| CULT | 1.877 | 1.215 | 0.095 | 161 | −6.533 | 0.000 |
| MGS | 2.759 | 2.085 | 0.164 | 161 | 1.583 | 0.942 |
| WAPP | 1.919 | 1.337 | 0.105 | 160 | −5.512 | 0.000 |
| DATA | 3.568 | 0.695 | 0.055 | 161 | 19.559 | 1.000 |
| ANLY | 3.398 | 1.219 | 0.096 | 161 | 9.381 | 1.000 |
| MEAN | 3.007 | 0.762 | 0.060 | 161 | 8.467 | 1.000 |

The last row of Table 7 shows mean statistics for all eleven dimensions, including a *t*-value of 8.467 with significance level greater than 0.05. It clearly shows that the mean of all eleven dimensions rating (3.007) is significantly higher than the threshold rating of 2.5, confirming that H1 is supported as above. Table 7 shows significant (*p*-value higher than 0.05) statistics for the dimensions compliance, competency, connectivity, collaboration, leadership, management system, data gathering, and analytics. The study respondents report that these dimensions are significantly implemented in their industries, whereas the results were not significant for culture, scalability, and app development, indicating limited industry implementation (*p*-value less than 0.05). These findings support the findings shown in Figure 9 and then discussed. They also confirm that H2 is not true in that there is not significant industry implementation of the eleven dimensions of the Quality 4.0 framework.

To assess the trueness of H3, Pearson's correlations and significance (*p*-value) between the implementation responses for the eleven dimensions were also calculated and are presented in Table 8. The correlations indicate positive, negative, or no relationships between any two dimensions.

**Table 8.** Pearson's Correlations and Significance between the Eleven Dimensions of Quality 4.0.

| | CMPL | CMPT | CONN | COLL | SCAL | LEAD | CULT | MGS | WAPP | DATA | ANLY |
|---|---|---|---|---|---|---|---|---|---|---|---|
| CMPL | 1 | 0.212 * | 0.225 * | −0.021 | 0.082 | 0.126 | −0.016 | 0.113 | −0.045 | 0.198 | 0.252 * |
| CMPT | 0.212 * | 1 | 0.548 * | 0.196 | 0.172 | 0.186 | 0.145 | 0.175 | 0.158 | 0.002 | 0.368 * |
| CONN | 0.225 * | 0.548 * | 1 | 0.101 | 0.221 * | 0.353 * | 0.170 | 0.205 * | 0.280 * | 0.089 | 0.448 * |
| COLL | −0.021 | 0.196 | 0.101 | 1 | 0.211 * | 0.256 * | 0.135 | 0.235 * | 0.154 | −0.025 | 0.232 * |
| SCAL | 0.082 | 0.172 | 0.221 * | 0.211 * | 1 | 0.261 * | 0.433 * | 0.607 * | 0.680 * | 0.057 | 0.467 * |
| LEAD | 0.126 | 0.186 | 0.353 * | 0.256 * | 0.261 * | 1 | 0.355 * | 0.423 * | 0.273 * | 0.098 | 0.455 * |
| CULT | −0.016 | 0.145 | 0.170 | 0.135 | 0.433 * | 0.355 * | 1 | 0.435 * | 0.571 * | 0.142 | 0.417 * |
| MGS | 0.113 | 0.175 | 0.205 * | 0.235 * | 0.607 * | 0.423 * | 0.435 * | 1 | 0.558 * | 0.065 | 0.485 * |
| WAPP | -0.045 | 0.158 | 0.280 * | 0.154 | 0.680 * | 0.273 * | 0.571 * | 0.558 * | 1 | 0.029 | 0.498 * |
| DATA | 0.198 | 0.002 | 0.089 | −0.025 | 0.057 | 0.098 | 0.142 | 0.065 | 0.029 | 1 | 0.182 |
| ANLY | 0.252 * | 0.368 * | 0.448 * | 0.232 * | 0.467 * | 0.455 * | 0.417 * | 0.485 * | 0.498 * | 0.182 | 1 |
| Correlation | 3 | 3 | 7 | 4 | 7 | 7 | 5 | 7 | 6 | 0 | 9 |

* significant correlation exists with 95% confidence interval.

Table 8 shows the Pearson correlation coefficients between the eleven dimensions, with significance (*p*-value less than 0.05) indicated by *. It can be seen from Table 8 that all positive coefficients were significant, showing that the correlations between these dimensions were direct. That is, if one axis is implemented in the industry, it will support the implementation of another axis. Hence, based on the correlation analysis, we can group the eleven dimensions of the Quality 4.0 based on their relationships to each other. It can be seen from the table that data gathering is the only axis having no relationship with any other axis. Analytics is the only axis having direct relationships with nine other dimensions (only excluding data). Connectivity, scalability, leadership and management system have direct relationships with the six other dimensions, whereas app development and culture have direct relationships with six and five other Quality 4.0 dimensions, respectively. Excluding data gathering, each of the eleven dimensions has a significant relationship with at least one of the Quality 4.0 dimensions, supporting H3.

H4 was tested regarding whether the Quality 4.0 framework is being well implemented in Pakistani multinational organizations, a sample size of 48. A target value of 4.0 (major implementation) was chosen as the determinant. *t*-test statistics were calculated for the "MEAN" of the mean scores for the eleven dimensions of Quality 4.0, which are presented in Table 9.

**Table 9.** *t*-test Statistics for Hypothesis H4.

|  | Mean | St. Dev. | SE of Mean | DOF | *t*-Value | Significance (*p*-Value) |
|---|---|---|---|---|---|---|
| MEAN | 3.373 | 0.713 | 0.103 | 47 | −6.090 | >0.001 |

The significance level (*p*-value) for the t-value of −6.090 is less than 0.05, which clearly shows that the null hypothesis of H4 is not valid. That is, the Quality 4.0 framework is not well implemented in the multinational organizations in Pakistan.

To determine the trueness of H5, data from the large-scale organizations (more than 500 employees) regarding the eleven dimensions of the Quality 4.0 framework, a sample size of 101. Since H5 refers to the well implementation of the Quality 4.0 framework, a mean score of 4.0 (major implementation) was set as the determinant. *t*-test statistics were calculated for the "MEAN" of the mean responses for the eleven dimensions. The results of the *t*-test are presented in Table 10.

**Table 10.** *t*-test for Hypothesis H5.

|  | Mean | St. Dev. | SE of Mean | DOF | *t*-Value | Significance (*p*-Value) |
|---|---|---|---|---|---|---|
| MEAN | 3.112 | 0.759 | 0.076 | 101 | −11.766 | >0.001 |

Again, the significance (*p*-value) is less than 0.05, which shows that the null hypothesis of H5 is also not valid. That is, the Quality 4.0 framework is not well implemented in large Pakistani companies of more than 500 employees.

To assess the validity of H6, one-way analysis of variance (ANOVA) was applied to the "MEAN" of the mean responses for the eleven dimensions with respect to the independent variable "NOQP", which is the number of quality professionals in the organization. The ANOVA results are presented in Table 11.

**Table 11.** ANOVA Results for the MEAN with respect to the NOQP.

|  | Sum of Squares | DOF | Mean Square | F-Value | Significance Level |
|---|---|---|---|---|---|
| Between Groups | 6.174 | 4 | 1.544 | 2.776 | 0.029 |
| Within Groups | 87.290 | 157 | 0.556 |  |  |
| Total | 93.464 | 161 |  |  |  |

The table shows that that the significance value is less than 0.05, which depicts that the factor NOQP, with an F-value of 2.776, significantly affect the implementation of the Quality 4.0 framework. Hence, the null hypothesis of H6 is true, which states that the implementation of Quality 4.0 framework improves with the number of quality professionals in any organization in an industry.

Finally, Table 12 presents the results and validation for all six hypotheses tested in this research.

**Table 12.** Summary of the Hypothesis Results.

| No. | Hypothesis | Test | Result | Remarks |
|---|---|---|---|---|
| H1 | The Quality 4.0 framework is implemented in the industry. | *t*-test Target value = 2.5 | True | Mean of means (MEAN) > 2.5 |
| H2 | All eleven dimensions of the Quality 4.0 framework are implemented in the industry. | *t*-test Target value = 2.5 | False | Not all means are greater than 2.5 |
| H3 | The eleven dimensions of the Quality 4.0 have relationships among each other. | Sig. Pearson Correlation Coefficient | True | Most of the dimensions are correlated |
| H4 | The Quality 4.0 framework is well implemented in multinational organizations. | *t*-test Target value = 4.0 | False | *p*-value < 0.05 |
| H5 | The Quality 4.0 framework is well implemented in large organizations having more than 500 employees. | *t*-test Target value = 4.0 | False | *p*-value < 0.05 |
| H6 | The existence of Quality Management professionals enables an organization to implement the Quality 4.0 framework | ANOVA F-value | True | NOQP is significant |

## 5. Recommendations

Based on the literature review, statistical analysis, and results, Table 13 presents recommendations for the implementation of the Quality 4.0 framework in Pakistani industries.

**Table 13.** Recommendations based on strengths, weaknesses, opportunities, and threats (SWOT).

| SWOT | S# | Recommendations |
|---|---|---|
| STRENGTHS | 1. | According to the respondents, COMPETENCY of human resources is one of the highest implemented dimensions of the Quality 4.0 framework, but the competency is not well utilized, resulting in low industry implementation of compliance, culture, scalability, and app development. CONNECTIVITY axis is very well implemented in Pakistani industries, but the industry infrastructure and the technology are quite obsolete. Industrial Internet of Things (IIoTs), high-speed internet, cloud computing, and app developments are necessary for industries to effectively integrate quality management systems. |
|  | 2. | As per the respondents, LEADERSHIP has the third highest implementation in industry in Pakistan. This dimension measures the involvement of the leadership in developing organizational strategy, vision, and mission; engagement with the quality management system, and implementation of Quality 4.0. This is for the obvious reason that the involvement of the leadership is critical because the outcomes and development of any organization are dependent on it. In this study, leaders demonstrated seriousness towards the implementation of Quality 4.0, which motivates the stakeholders in implementing the new concepts of Quality 4.0 in industries. |

**Table 13.** *Cont.*

| SWOT | S# | Recommendations |
|---|---|---|
| WEAKNESSES | 1. | Fostering a quality CULTURE is the first essential element in developing and implementing Quality 4.0. Technological advancement to achieve breakthrough performance indicators can only transform the organization when employees and other stakeholders from the top down take ownership of quality. Quality Culture and its sustainability is one of the major issues in Pakistani industries, reflected in the low respondent scores. It is recommended that leadership and human resources staff change their mind-sets towards quality management systems to enhance the customer experience and achieve continuous sustainability. Quality culture can be developed in organizations through rewards and recognition systems, especially through the Kaizen approach recommended under the globally famous Toyota Production System/Lean. It is strongly recommended that organizations start with quality principles/practices and then implement the basic tradition tools/techniques in moving towards Quality 4.0. |
| | 2. | The Quality 4.0 framework is quite technology oriented, which requires technological infrastructure and skilled and trained human resources to make continuous updates, which requires huge resources for any organization. Hence, determining the costs of quality analysis (ANALYTICS) is recommended for the implementation [39]. |
| | 3. | Technology and training updates are the integrated feature of the Quality 4.0 framework; hence it is recommended to acquire the human resources (COMPETENCY) willing to continuously adopt the changes in technology and trainings for the continuous improvement of systems. |
| | 4. | It is recommended to provide industry awareness sessions especially in the third world countries developing countries to disseminate the features, importance, and advantages of implementing Quality 4.0. |
| | 5. | The COMPLIANCE and standardization at the organization and industry levels must aim at enhancing in maturity and customer experience. |
| | 6. | The COMPETENCY of the human resources may be effectively utilized for the scalability of the organization to cope with the fluctuating demands. The major dilemma in industries is the constant production rates and capacity, resulting in the low utilization of resources in non-peak demand periods. |
| | 7. | The existence of QUALITY MANAGEMENT SYSTEMS (QMS) is quite weak. They are implemented in the industry mostly to comply with ISO 9001 standards and surveillance audits. The QMS It is recommended that both the manufacturing and service industries adopt a QMS and evaluate it through the internal and surveillance audit. |
| | 8. | Integration and CONNECTIVITY of the real-time data from the IIoTs, POS, and other sources may be collected through cloud-based data collection systems, which may be eventually analysed, forecast, and displayed on dashboards. |
| | 9. | There was a considerable lack of data analysis (ANALYTICS); data were often gathered for the sake of auditing purposes. Data analysis is strongly recommended whether in traditional ways or towards Quality 4.0. |
| OPPORTUNITIES | 1. | LEADERSHIP capabilities can be exploited for the improvement in Quality Management by having frequent review meetings, dashboard discussion, customer reviews, etc. Leadership implementation and execution of Quality 4.0 can enhance overall quality and the customer experience. Unlike the Total Quality Management philosophy (CULTURE), Quality 4.0 is an implementable framework that can be adopted by the manufacturing and service industries for enhanced productivity and customer experience. |
| | 2. | The Quality 4.0 framework is as perceived in the previous research not only is an operational framework but also is directly linked with the strategy of any organization (MANAGEMENT SYSTEM). Strategic planning, execution, implementation, and evaluation are integral to the Quality 4.0 framework. |
| | 3. | Since the framework relies on COLLABORATION and input from the government, industry, and other stakeholder, it is recommended for the smooth implementation of the framework that the framework be implemented in stable organizations. This is because volatility and continuous changes in stakeholder requirement may result in the ineffective implementation of the framework, resulting in the loss of resources. |
| | 4. | It is recommended to develop and implement smart KPIs (DATA) and benchmarks for the measurement of performance at both the macro and micro levels. Real-time collection and analysis of data for KPI dashboards (APP DEVELOPMENT) are highly recommended to be available for review through the leadership in the industry. |
| | 5. | AI-based cloud computing, which is an important contributor SCALABILITY and decision systems (DATA), are integrated into the Quality 4.0 framework for data analysis and inferences, and results may be communicated to the relevant people. |
| | 6. | In this era of big data collection and analysis, it is important for human resources to be aware of the importance of real-time data collection and analysis. |
| | 7. | To oblige the compliance axis of the Quality 4.0 framework, it is recommended that at the government and industry levels, international benchmarks and COLLABORATION be introduced to enhance industry competitiveness. |

**Table 13.** *Cont.*

| SWOT | S# | Recommendations |
|---|---|---|
| THREATS | 1. | LEADERSHIP must make great investments to transform. |
| | 2. | Due to the unavailability of 5G technology (CONNECTIVITY) in the third world developing countries, cloud computing (SCALABILITY) and smart decision making are huge problems, resulting in high scrap and defect rates in industries. |
| | 3. | Most technology and equipment specified in the general framework of Quality 4.0 are not designed and manufactured in developing countries, and thus, their acquisition can be a hurdle; in addition, there can be delays not connected to industry targets due to frequent changes in government policies (COMPLIANCE), especially in Pakistan. |
| | 4. | In consideration of digital transformations, traditional quality practices are recommended to drawing attention to sustainability. A lack of long-term proper application of traditional quality practices, especially in small to medium-scale industries was observed. Employees feel overburdened by following specified mandatory quality practices and have the perception of adding waste (cost) instead of adding quality. Quality CULTURE is recommended. |
| | 5. | Employees feel threats of downsizing following technological advancements; they must be counselled regarding the benefits of the advancement of quality CULTURE by leadership. |

## 6. Conclusions

In this research, a framework for the implementation of Quality 4.0 in an enterprise has been proposed. The framework proposed in this research is based on the philosophy of Quality 4.0 presented by Juran, although an implementable framework has been suggested in the proposed work. Based on the developed Quality 4.0 framework, the maturity of Pakistani industry was determined. A comprehensive questionnaire comprising of 56 questions was developed that linked to the eleven dimensions of the Quality 4.0 framework. Survey responses were gathered and analysed from 161 quality practitioners and professionals from different industries. Based on the questionnaire responses, means were calculated for each of the eleven dimensions are determined, and a final mean provides the overall status of the industry.

Based on the statistical analysis of the respondents' data, the following comments can be made:

- Quality 4.0 is not fully implemented in Pakistani industry.
- The eleven dimensions of the Quality 4.0 are not implemented well in the industry, especially compliance, scalability, culture, and app development. These need the attention of industry for implementation.
- Implementation of one dimension of the Quality 4.0 framework has direct impacts on the others; hence, complying with the requirements of one dimension may result in the implementation of other Quality 4.0 dimensions.
- Quality 4.0 is not implemented in small domestic organizations, but more in-depth analysis shows that it is also not well implemented in multinational and large organizations.

The number of respondents and sample size can be increased to improve these results. Future research can be extended to developed and other developing countries. Furthermore, the implementation of the developed framework may be assessed for its effectiveness in different industry sectors such as finance or service. The framework was developed to be generic, and its strengths and weaknesses need to be measured in different manufacturing and service sectors so that it can be improved. The framework may also be modified for the specific needs of industries, especially for SMEs.

Initially, the sustainability of traditional quality practices must be properly planned and executed for long-term digital transformations from traditional quality. Later, the maturity of cycles of continuous sustainability can be evaluated specifically for the agriculture, industrial, and services sectors, the major contributors to the GDP of Pakistan [40].

**Author Contributions:** Conceptualization, A.Z.; methodology, A.Z.; software, M.W.; validation, A.Z. and S.A.I.; formal analysis, M.W.; investigation, A.Z. and M.W.; resources, M.W.; data curation, A.Z.; writing—original draft preparation, A.Z. and M.W.; writing—review and editing, A.Z. and S.A.I.; visualization, A.Z.; supervision, S.A.I.; project administration, M.W. All authors have read and agreed to the published version of the manuscript.

**Funding:** This research received no external funding.

**Institutional Review Board Statement:** Not applicable.

**Informed Consent Statement:** Informed consent was obtained from all subjects involved in the study.

**Data Availability Statement:** Not applicable.

**Acknowledgments:** The authors are highly grateful to the industry respondents who participated in this research. We also acknowledge the authors of the articles we referred to in the research.

**Conflicts of Interest:** The authors declare no conflict of interest.

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
