# Peer review of "Developing a Quality 4.0 Implementation Framework and Evaluating the Maturity Levels of Industries in Developing Countries"

_sustainability, doi:10.3390/su141811298_

Round 1

Reviewer 1 Report

Manuscript ID sustainability - 1853468

Title: «Development of Quality 4.0 implementation framework and Evaluation of the Maturity Level for the Industries in developing countries».

Considering the current trends in the development of quality management, the presented manuscript is relevant from a scientific and practical point of view. Authors propose a framework for the implementation of Quality 4.0 in the industry. This framework is based on its eleven dimensions or axes, which are the core requirements of the Quality 4.0. For the implementation of framework, key variables are evaluated to gauge the maturity of implementation of the framework. As a whole, methodological inaccuracies are not detected.

This manuscript includes the next harmonious structure: 1. Introduction (p. 1 – 5); 2. Conceptualization of Quality 4.0 Framework (p. 5 – 8); 3. Research Instrument & Sampling (p. 8 – 11); 4. Statistical Analysis & Results (p. 11 – 18); 5. Recommendations (p. 18 – 19), 6. Conclusion (p. 19 – 20). The figures and tables are appropriate, they properly show the data. So, they easy to interpret and understand by readers. References list is adequate and includes 29 titles.

The value of this work is significant.

Reviewer 2 Report

1) Industry 5.0 is not covered at all in this article. In deliberations and summing up, it exposes Quality Management (QM) only from the technology side. This results in a lack of actual analysis (or even falsehood) of the discussion of results, e.g. in the Culture or Leadership axis.

2) The Recommendation part (e.g. but the infrastructure and the technology used in the industry is quite obsolete.)goes far beyond the content of the article which is focused on numbers and statistics

3) Completely no conclusions or practical recommendations regarding the implementation of the Q 4.0 Framework.

4) The article proposes only a framework with 11 axes. It is quite a complicated methodology and requires quite a complex project in the organization. Whether? When? and How? it is possible to carry out a simplified framework, in accordance with the requirements, priorities or capabilities of an organization, e.g. SME?

(very minor) detailed comments

·       line 66 QM is definitely not CORE to every business. QM is maybe common, or usuall.

·       line 77, the concept of standardization and standard should be explained in the understanding of Agile methodologies and in Knowledge Economy,

·       line 107-109 It is necessary to mention Industry 5.0 and the requirements of knowledge-intensive business processes (kiBPs),

·       line 192 The drawing is incomprehensible to practitioners. The content of the work does not present in any way how the Q4.0 Framework is / should be implemented or processed.

·       line 204 and 2017 versus legal requirements? versus environmental requirements?

·       line 252-253 The content shows that the strategy is the result of SWOT QM analysis? Which of course is not true.

·       line 272 -284 Industry 5.0 (!) -clearly lacks attention. The whole thing strongly is from the previous era and does not refer to Industry 4.0 / 5.0 or Knowledge Economy.

·       line 292-293 And unstructured data ?? Do we limit QM only to mass Production?

·       line 311 -321 Web Apps -a very bad, misleading name, denoting a great many different technologies nowadays. Rather different name "Information System" or "Information flow system".

·       line 323-333 And how will it be in services? Do we limit QM only to mass Production?

·       Recommendation Weknessess:

o   p.1 It is worth mentioning Industry 5.0 and empowerment or maybe dynamic BPM and kiBPs.

o   p. 2 Exactly as above.

Round 2

Reviewer 2 Report

Corrected article still:

·       misrepresents the concept of Industry 5.0, ignoring the role and importance of an EMPLOYEE, a person performing work using as many opportunities as possible for improvement, innovation and development, also personal, thanks to the entitlement to non-standard activities (! Empowerment, dynamic BPM). (“Majorly the Industry 5.0 enhanced the customer experience and satisfaction in terms of real-time traceability, pro-active customer care and customer involvement in designing the product and services”).

·       Section 2.5 Collaboration is still from the previous era and does not refer to Industry 5.0 or Knowledge Economy (KE) etc.

·       In the summary, the authors only mention the possibility of running the framework in a simplified version, without giving any practical recommendations. What really prevents the use of the framework in SME.
